# Social and Financial Inclusion through Nonbanking Institutions: A Model for Rural Romania

**Xiao-Guang Yue** [1,2]📧, **Yong Cao** [3], **Nelson Duarte** [4]📧, **Xue-Feng Shao** [5] and **Otilia Manta** [6,7,*]📧

1    Rattanakosin International College of Creative Entrepreneurship, Rajamangala University of Technology
     Rattanakosin, Nakon Patom 73170, Thailand; xgyue@foxmail.com
2    School of Sciences, European University Cyprus, Nicosia 1516, Cyprus
3    Research Institute of Big Data and Artificial Intelligence, Southwest Forestry University, Kunming 650224,
     China; xgyue@whut.edu.cn
4    CIICESI, School of Management and Technology, Porto Polytechnic, 4610-156 Felgueiras, Portugal;
     nduarte@estg.ipp.pt
5    Business School, La Trobe University Sydney Campus, Sydney 2000, Australia; x.shao@latrobe.edu.au
6    Romanian Academy, Center for Financial and Monetary Research—Victor Slăvescu,
     050711 Bucharest, Romania
7    Romanian-American University, 012101 Bucharest, Romania
*    Correspondence: otilia.manta@rgic.ro or otilia.manta@icfm.ro; Tel.: +40-722-614-031

**Abstract:** The challenges of financial systems have immediate or medium-term social effects. The financial industry is constantly searching for measures to reduce these challenges, especially for those with little or no access to financial services. While current communication technologies make services more accessible through digital mobile platforms, there are still difficulties in establishing viable customer arrangements. In addition to the increased investment in financial technologies, nonbanking financial institutions have now expanded to offer more flexible services tailored to individual circumstances, especially those in isolated rural areas. This research outlines the network model of nonbanking financial institutions in Romania, as well as a microfinance model, based on the financial analysis of four national indicators of nonbanking financial institutions. Data used are presented in absolute values, from the annual numerical series for the reference period 2007–2017. The new initiatives and features incorporated in this Romanian model should be applicable elsewhere and will actively contribute to the expansion and sustainability of financial services, with a positive inclusive impact on society.

**Keywords:** community finances; fiscal flexibility; individualized financial arrangements; sustainable financial services

## 1. Introduction

In a global context where large populations have inadequate access to financial services and slow local and regional development, this study assesses the expansion of the nonbanking financial institution (NFI) sector in recent years; it also briefly reviews innovative financial models designed to improve access to and the relevance of financial services for people previously not participating in regulated financial systems. These investigations give indications of successful ways to promote social inclusion and its benefits.

With the emergence of digital financial technologies and the societal challenges created by globalization, it is essential to develop the economic models that improve economic, social, and financial

inclusion (Bartels 2017; Shaikh et al. 2017; Nanda and Kaur 2016). Digitization has had an extraordinary impact on the development of financial innovations, especially in the fields of digital finance and financial institutions in virtual space, namely, the so-called "virtual banks" (Pennathur 2001).

Artificial intelligence is another major development in the financial field. This is a natural trend because many jobs in the financial banking system are repetitive in nature, and robotic components have already been widely implemented (e.g., processing/transfer of payments online or via ATM, account opening/closure). At the same time, the financial technology (FinTech) concept has created new opportunities for clients, such as rapid identification of individual businesses and high flexibility of the award conditions (Kuo Chuen and Teo 2015). This financial revolution has expanded the context of microfinance and provided digital opportunities for the next generation (Arp 2018; Arp et al. 2017). The emergence of new systems such as GlobTech—an innovation of global economic and financial technologies, as well as FinTech—blockchain technology has stimulated confidence in new concepts and financial products (Manta 2017). Financial regulators are also tightening measures against money laundering and terrorism financing.

## 2. Literature Review

The current financial reality is the assessment of the adaptation of current financial systems to new financial technological trends. In general, the new financial reality is reflected in new business concepts, such as FinTech. In order to adapt to these challenges, it is necessary for users to know the main major technological trends, including cryptocurrencies, artificial intelligence, blockchain, and database management, from both the business and regulatory points of view. We should also understand how to analyze and evaluate technological innovation in finance, and how new technologies impact economies, markets, companies, and individuals (Sanicola 2017; Zhang et al. 2019).

The existing literature uses the term FinTech, which is generally defined as "a new financial industry that applies technology to improve financial activities" (Schüffel 2017; Government and Economy 2018). FinTech introduces new approaches to financing applications, processes, products, and business finance; it emerged from the need to optimize the funding process through technology and is made up of one or more complementary financial services, provided as an end-to-end process via the Internet (Australian Government 2010; BRW 2014; Sanicola 2017). Financial technology has been used to automate insurance, transactions, banking, and risk management (Aldridge and Krawciw 2017; Wesley-James et al. 2015). Financial services in the process of social and financial inclusion can be provided by innovative financial service providers, or may come from an authorized banking institution or even an insurer. The penetration of financial services to the beneficiary is currently possible through online platforms based on financial-banking programming applications, applications for the financial sector and are supported by regulations such as the European Payment Directive (Scholten 2016).

In trading on the capital markets, innovative electronic trading platforms have emerged as a result of the optimization of the financing process both in terms of time and the operational management of financial transactions in the virtual space. Social trading networks allow investors to observe the trading behavior of traders and to follow their investment strategies on financial markets (Voronkova and Bohl 2005). The platforms require little knowledge about financial markets and have been described as being "a cheap, sophisticated alternative to traditional wealth managers", according to the World Economic Forum (McWaters 2015). However, given the standardization of repetitive activities, a new class of robocounsellors has emerged. This class is a group of automated financial advisers, who provide financial advice to manage online investment with minimal to moderate investment (Lieber 2014; Redheering 2016).

In less than a decade, global investment in financial technology has increased by more than 2200%, from $930 million in 2008 to more than $22 billion in 2015 (Accenture 2019). The financial technology industry has also seen rapid growth in recent years. According to the mayor's office in London, 40% of London's workforce is engaged in financial and technological services (Accenture 2019). As a leading

global professional services company, Accenture analyzed data obtained from a global venture-finance data and analytics firm—CB Insights—and found that global investments in FinTech increased to $55.3 billion in 2018, with the investments in China taking the lead. The total value of transactions doubled, with the number of transactions being close to 20%. Sales values in the United States and the United Kingdom increased by 46% and 56%, respectively; Canada, Australia, Japan, and Brazil also have fundraising (Accenture 2019).

The significant increase was largely due to a ninefold increase in the value of transactions in China to $25.5 billion—nearly as much as $26.7 billion of all FinTech investments globally in 2017. China accounted for 46% of all FinTech investments in 2018. More than half of China's FinTech investments came from the $14 billion financing round in May of Ant Financial, which manages the largest money market fund in world but is best known for its Alipay mobile payment service. In Europe, $1.5 billion was invested in financial technology companies in 2014. Specifically, London-based companies received $539 million, Amsterdam-based companies received $306 million, and companies in Stockholm received $266 million. FinTech's transactions in Europe reached a maximum of five quarters, rising from 37 in the fourth quarter of 2015 to 47 in the first quarter of 2016 (Wesley-James et al. 2015). Lithuania is becoming a Northern European center of financial technology companies, with the exit of Great Britain from the European Union, and has issued 51 FinTech licenses since 2016 and 32 of them in 2017 (Government and Economy 2018; Financial Stability Board 2019). FinTech companies in the United States raised $12.4 billion in 2018, showing an increase of 43% compared with 2017 figures (Kauflin 2019).

The idea of "financial inclusion" has gained acceptance and importance since the early 2000s, when exclusion from financial services was identified to be directly correlated with poverty (Armenion 2016; Scholten 2016; Sarma and Pais 2011). In 2018, it was estimated that over 2.2 billion working-age adults, globally, did not have access to financial services provided by regulated financial institutions. For example, in Sub-Saharan Africa, only 24% of adults have a bank account, even though the official African financial sector has grown in recent years (Sanicola 2017). The United Nations (UN) defines financial inclusion objectives (McWaters 2015) as the access to a reasonable cost for all households of a complete range of financial services, including savings or storage, payment, and credit transfer and insurance services. The UN, through partnerships with financial institutions, supports the phenomenon of financial inclusion of the many in need. It adapted and developed personalized financial products for the poor, as well as promoted integrated financial education programs regarding innovative financial services and products, which strengthen the knowledge of financial services through the financial education process, in particular by involving women. The UN financial inclusion product is funded through the United Nations Development Program (Accenture 2019).

Financial exclusion is a big issue of social exclusion in a society, which prevents individuals or social groups from gaining access to formal financial system (Sarma 2008). Those who promote financial inclusion argue that financial services have positive effects when more investors and businesses are involved (Chakelian 2016; UNDP 2012; Williams-Grut 2015). This is also confirmed by the regulatory policy of NFIs that support financial products and services aimed directly at supporting social inclusion. With the direct involvement of social inclusion and a sense of belonging in their community, firms can benefit in different ways. For example, Dunbar et al. (2019) found that enterprises open to corporate social responsibility (CSR) are likely to bring reputational intangibles with risk-reduction incentives at the managerial level (Dunbar et al. 2019; Liu et al. 2017). However, there is also skepticism about the effectiveness of financial inclusion initiatives (Schüffel 2017). Research on microfinance initiatives indicates that broad credit availability for microenterprises can produce informal mediation, a form of unintended entrepreneurship (Aldridge and Krawciw 2017). Nevertheless, from a contingency perspective, firms undertaking CSR activities witness more investment opportunities and stronger governance secures its financial services in an all-inclusive financial system, which, in return, improves future social responsibilities (Ikram et al. 2019).

Clearly, at a global level, NFIs play an important role for nonbank customers. Through innovative financing instruments, the financial regulations regarding their activity, and the programs and funding models, nonbanking financial institutes will be better supporting those who are financially excluded, for instance, small and marginal farmers and certain social groups (Sunstar Philippines 2016). In Romania, an open approach to digital financial technologies exists, but several steps are necessary before these new technologies can be adopted. The first step is to promote knowledge of FinTech advantages. This research provides further analysis of the network of nonbanking financial services, as well as indicators of its development in Romania. By using a case study, this paper identifies specific assumptions and criteria for the design and development of a new microfinance models for rural areas.

## 3. Methodology

### 3.1. Descriptive Data and Statistics

In an increasingly dynamic financial services industry, banking institutions, NFIs, and other digital financial institutions (FinTech) form the essential parts of a solid and stable financial system. These financial institutions were initially established in different financial sectors. However, they need to be merged into a well-coordinated and complementary system. In some countries, the banking system dominates, whereas in others, nonbank financial institutions (including digital ones) are creating an alternative complement to the banking system. This creates easier access to finance for businesses and households. A range of financial products and services are currently provided both by banks and nonbank financial institutions, as well as many other organizations, including insurance, leasing, factoring, venture capital companies, mutual funds, and pension funds. The ratio of stock market capitalization to banking system assets is very high in most economically advanced countries.

In order to meet the challenge in national financial systems, we constructed the NFI indicators of Romania for 2007–2017 (Ministry of Public Finance 2018; National Institute of Statistics 2014). Data sources include Romania's statistical yearbook, national accounts, national financial accounts, National Bank of Romania statistics, and the financial statements of NFIs and the Financial Supervisory Authority (FSA). The absence of complete NFI data required the use of correlation coefficients between the financial sectors and subsectors: investment funds other than money market funds, other financial intermediaries excluding companies' insurance and pension funds, and financial auxiliaries.

### 3.2. Indicators and Models

Explanations regarding the indicators used in our research are summarized below in Table 1.

**Table 1.** Parameters of the four indicators used in the model.

|  | Name | Symbol | Data Source | Calculation Formula |
|---|---|---|---|---|
| **Indicator 1** | Asset formation indicator | IF | Nonbanking financial institution (NFI) financial statements and national financial accounts | IF = CHN/AFT = total financial expense/assets |
| **Indicator 2** | Asset usage indicator | IU | NFI financial statements and national financial accounts | IU = VTN/AFT = total income/total financial assets |
| **Indicator 3** | NFI sector expansion indicator | IS | Financial statements of the NFIs and the statistical yearbook of Romania | IS = AFN/GDP = total financial assets/GDP |
| **Indicator 4** | Debt sustainability indicator | IT | NFI financial statements and national financial accounts | IT = DFN/VTN = total financial debt/total income |

The asset formation indicator (IF) highlights the financial effort of an NFI to set up total assets and, in particular, financial assets and can help differentiate between organizations in asset formation. It allows for assessment of the effectiveness of a policy for mobilizing financial resources and management of their allocation by type of financial assets (investments) in an economic analysis. In this study, IF

created possibilities for correlation with performance indicators, development, as well as openings for interpretation of the penetration potential in the financial markets specific to NFIs.

The asset usage indicator (IU), also referred to as the asset rotation indicator, is both a measure of the economic profitability of NFIs and the efficiency of asset use, independently of the financial structure, the fiscal policy that taxes the profit, as well as the exceptional elements. IU highlights the globally active, generative relationship between the balance sheet and the results, a relationship dependent on the set of internal and external factors influencing the activity of NFIs, including the managerial factor. It provides a possibility of a quantitative approach to the interactivity between the financial variables of NFIs, the elements of the active patrimony, and the elements of the "total" productive results. This indicator could be useful for building a synthetic financial assessment indicator of NFIs.

The NFI sector expansion indicator (IS) highlights the importance of the NFI sector in the economy and its share, through inclusion, in the financial sector in terms of financial potential. It allows knowledge of the extension of the NFI sector, offering the possibility of evaluating the sector's connection to the national economy as a whole by processing additional information to the financial sector components of the financial markets. This indicator has its advantages in economic analysis with the possibilities for correlation with efficiency, performance, robustness, and stability indicators. The NFI sector expansion indicator provides indirect information on the interconnection between the result component (efficiency and performance) and the potential (robustness and stability) component of the NFI sector. Matrix integration of the indicator in the network of interactions between the evaluation indicators can be executed for the purpose of composing modeling of their dynamic codeterminations and the construction of a synthetic indicator of the financial evaluation of NFIs.

The debt sustainability indicator (IT) expresses the ability of NFIs to generically secure the liquidity of the debt from the revenue generated by the financial investments, expressing indirectly the indebtedness possibilities of the NFI and, at the same time, its margin of maneuver for the structuring of financial debts. It allows for a correlated analysis of financial leverage and performance indicators and debt recovery, providing information on the potential for improving the market position of NFIs.

### 3.2.1. Credit Constraints Model (Model 1)

In this subsection, we propose four indicators to evaluate financial system developments. Our choice of the indicators is in accordance with the literature and also takes into account the fact that microfinance and other financial innovations are still in their infancy in Romania. The input variables were obtained by generalizing and integrating items in the balance sheet and NFI result account, representing the state and, sometimes, dynamics of financial aggregates. The indicators were calculated from primary indicators and represent absolute values of input variables. Relative weights of the calculated indicators are determined on the basis of the logical conditioning, causal correspondence, absolute values of input variables, or comparability of data in terms of content, coverage, processing methodology, units of measure, and sources of information. The primary and calculated indicators used in this paper are expressed in absolute terms and listed in Table 2.

**Table 2.** Terms of primary and calculated indicators.

| Primary Indicators | Terms |
|---|---|
| AFN | Total financial assets of NFI, comprising total investments, fixed assets, disponible assets, and receivables |
| CHN | Total NFI expenditure, comprising total financial and nonfinancial expenditure |
| GDP | Gross domestic product. It is found that this indicator is determined by comparing two absolute sizes of different natures—a stock size and a flow rate |
| VTN | Total NFI revenue |
| DFN | Total NFI financial liabilities |
| **Efficiency (EF)** | |
| IF | Asset formation indicator = total expenditure/total financial assets = CHN/AFN |
| **Performance (PF)** | |
| UI | Asset Use Indicator = total income/fine assets total = VTN/AFN |
| **Development (DV)** | |
| IS | NFI growth indicator = fine assets total/GDP = AFN/GDP |
| **Sustainability (ST)** | |
| IT | Debt sustainability indicator = total financial debt/total revenue = DFN/VTN |

### 3.2.2. Network Model of Nonbank Financial Institutions (Including Digital Ones)

To investigate the flexibility of the nonbanking financial system, the proposed model intuitively represents a real financial system that retains the essential features relevant to modeling its impact. According to basic exogenous parameters, the simulation must represent the financial system, allow study of the flexibility of the financial system to shocks, and indicate how shock flexibility depends on the key parameters of the system.

The structure of the financial system is based on two exogenous parameters describing the random graph: the number of nodes representing nonbanking/digital financial institutions ($N$) and the probability ($p_{ij}$), where i is one NFI that is linked to another NFI j and the probability is assumed equal for all pairs of nonbanking/digital financial institutions. The graph simulation was done according to the specified parameters and shows a number of links Z made in Figure 1, with a significant network of nine NFIs.

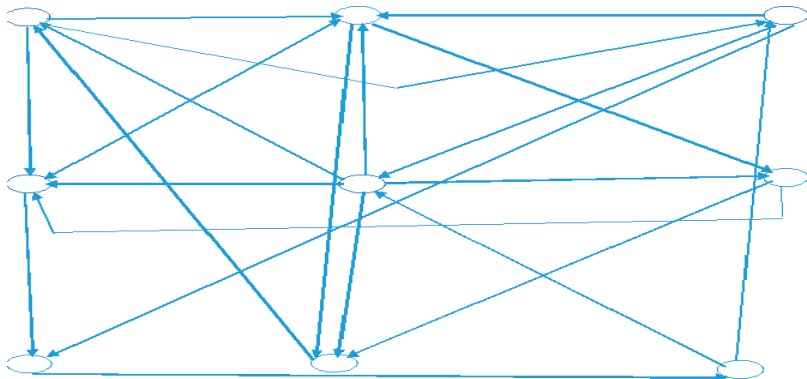

**Figure 1.** The representation of the network of nine NFIs. Source: Otilia Manta, based on her own contribution.

For any representation of the random graph, the balance sheet of individual nonbank financial institutions was completed in a consistent manner with the level of the financial institution and the

aggregate balance sheet identity. For the purpose of the detailed description, the following notation was inserted for clarity. The small letters are used for variables at the level of the individual NFI, the big letters for the aggregates, and the Greek letters for the rates.

An individual NFI asset, denoted by $a$, includes external assets denoted by $e$ and interbank assets denoted by $i$. Thus, for NFI, we have $a_i = e_i + i_i$, where $i = 1, \ldots, N$.

An NFI's liabilities, denoted by $l$, consist of the net assets of the NFI denoted by $c$, customer deposits denoted by $d$, and borrowings between financial institutions denoted by $b$. Thus, for NFIs, we have $l_i = c_i + d_i + b_i$, where $i = 1, \ldots, N$. According to the NFI balance, we have $a_i = l_i$, for $i = 1, \ldots, N$.

The asset side of the NFI balance sheet as well as the interinstitutional/interbank borrowing received ($b$) on the liabilities side are then added. The determination of the two remaining components, the net asset ($c$) and the deposits ($d$) on the liabilities side, is relatively direct. The net asset is established as a fixed proportion ($c$) of the total assets at the level of the NFI, $c_i = \gamma \times a_i$. Customer deposits are deducted as the rest of the identity of the NFI balance (i.e., $d_i = a_i - c_i - b_i$).

The network model of nonbank financial institutions, as well as each component of the NFI balance sheet, completes the construction of the system of nonbank financial institutions, supporting financial inclusion at national and global levels. The financial systems constituted by NFIs can be simply described by the following set of architectural parameters ($c$, $p$, $N$, and $E$), where $c$ signifies the net asset as a percentage of total assets, $p$ is the probability of any two nodes or financial institutions to be connected, $N$ is the number of NFIs, and $E$ is the total of external assets of financial institutions.

Another financing model that is complementary to the network model is the microfinance model (Manta 2018). The construction of the microfinance model described here was based on research tools such as interviews and questionnaires of over 15,000 nonbank users in rural Romania. The research was carried out in eight designated development regions in 2012–2016. The work included testing the microfinance model as a basis for social and financial inclusion, as well as facilitating entrepreneurship in rural areas of Romanian.

## 4. Results

In this section, we present the situation of the main indicators of the NFIs, as well as a microfinance model developed from a conceptual and testable point of view in the rural area for the Romanian market.

### 4.1. Empirical Evaluation of NFI (Calculation of Indicators)

The calculation of NFI values was carried out for the period 2007–2017, for which consistent data are available in Table 3. Sources of data used have been previously listed; however, in some cases, the absence of complete NFI data required the use of correlation coefficients, as explained at the end of Section 3.1.

**Table 3.** Absolute values of primary indicators (million RON[1]).

| Year | 2007 | 2008 | 2009 | 2010 | 2011 | 2012 | 2013 | 2014 | 2015 | 2016 | 2017 |
|---|---|---|---|---|---|---|---|---|---|---|---|
| Income (VTN) | 2756 | 2689 | 3002 | 6206 | 5880 | 5034 | 4882 | 5168 | 4272 | 4382 | 4955 |
| Costs (CHN) | 1617 | 2350 | 1997 | 6120 | 5665 | 4770 | 4748 | 4775 | 3908 | 3913 | 4452 |
| Total financial assets (AFN) | 29,766 | 42,565 | 36,375 | 30,850 | 27,221 | 26,772 | 25,927 | 24,020 | 25,235 | 28,084 | 32,476 |
| Liability (DFN) | 48,410 | 43,502 | 27,216 | 22,527 | 12,300 | 24,189 | 2402 | 23,848 | 22,404 | 23,311 | 24,920 |
| GDP (PIB) | 416,007 | 524,389 | 510,523 | 533,881 | 565,097 | 596,682 | 637,456 | 668,144 | 712,659 | 762,342 | 858,333 |

Source: values based on financial statements of NFI and national financial accounts for 2007–2017 (Ministry of Public Finance).

---

[1]　RON is the national currency of Romania; the average annual rate for 2019 is 1 Euro to 4.7383 RON, according to the National Bank of Romania.

### 4.1.1. Relevant Indicators for NFI Sector Development

Indicator 1: Asset Formation Indicator (IF)

Formula:

$$IF = CHN/AFN = \text{total financial expense/assets}$$

Values of the asset formation indicator (IF) were calculated and are reported in Table 4.

**Table 4.** Asset formation indicator (IF) values for the period 2007–2017.

| Year | 2007 | 2008 | 2009 | 2010 | 2011 | 2012 | 2013 | 2014 | 2015 | 2016 | 2017 |
|---|---|---|---|---|---|---|---|---|---|---|---|
| Costs (CHN) (million RON) | 1617 | 2350 | 1997 | 6120 | 5665 | 4770 | 4748 | 4775 | 3908 | 3913 | 4452 |
| Total financial assets (AFN), (million RON) | 29,766 | 42,565 | 36,375 | 30,850 | 27,221 | 26,772 | 25,927 | 24,020 | 25,235 | 28,084 | 32,476 |
| IF | 0.054 | 0.055 | 0.549 | 0.198 | 0.208 | 0.178 | 0.183 | 0.198 | 0.155 | 0.139 | 0.137 |

Source: values based on financial statements of NFI and national financial accounts for 2007–2017 (Ministry of Public Finance).

For the analyzed period, the increased efforts of the NFIs were demonstrated in the expenditures incurred and the constitution of the financial assets, showing that the expense for setting up a unit of financial asset had increased. This highlights the effects of the financial crisis, which resulted in increased competition in the NFI product market, as well as ineffective policy for mobilizing financial resources and allocating them. The crisis not only resulted in the collapse of financial institutions but also impeded global credit markets and required intensive government interventions in NFIs (Li et al. 2018). This trend impacted the performance of NFIs and their development, robustness, and stability.

Indicator 2: Asset Use Indicator (IU)

Formula:

$$IU = VTN/AFN = \text{total income/total financial assets}$$

Values of the asset user indicator (IU) were calculated and are reported in Table 5.

**Table 5.** Asset user indicator (IU) values for the period 2007–2017.

| Year | 2007 | 2008 | 2009 | 2010 | 2011 | 2012 | 2013 | 2014 | 2015 | 2016 | 2017 |
|---|---|---|---|---|---|---|---|---|---|---|---|
| Income (VTN) (million RON) | 2756 | 2689 | 3002 | 6206 | 5880 | 5034 | 4882 | 5168 | 4272 | 4382 | 4955 |
| Total financial assets (AFN), (million RON) | 29,766 | 42,565 | 36,375 | 30,850 | 27,221 | 26,772 | 25,927 | 24,020 | 25,235 | 28,084 | 32,476 |
| IU | 0.092 | 0.063 | 0.083 | 0.201 | 0.216 | 0.188 | 0.188 | 0.217 | 0.169 | 0.156 | 0.153 |

Source: values based on financial statements of NFI and national financial accounts for 2007–2017 (Ministry of Public Finance).

The evolution of this performance indicator reflects both asset efficiency and global economic efficiency. Its evolution was influenced by internal and external factors of NFI, especially the managerial factor. The low deterioration of the indicator's value is due to the offsetting effects of relevant indicators and the complementary impact of heritage elements in the NFI portfolio.

Indicator 3: NFI (IS) Expansion Indicator

Formula:

$$IS = AFN/GDP = \text{total financial assets/GDP}$$

Values of the NFI expansion indicator (IS) were calculated and are reported in Table 6.

**Table 6.** NFI expansion indicator (IS) values for the period 2007–2017.

| Year | 2007 | 2008 | 2009 | 2010 | 2011 | 2012 | 2013 | 2014 | 2015 | 2016 | 2017 |
|---|---|---|---|---|---|---|---|---|---|---|---|
| Total financial assets (AFN) (million RON) | 29,766 | 42,565 | 36,375 | 30,850 | 27,221 | 26,772 | 25,927 | 24,020 | 25,235 | 28,084 | 32,476 |
| GDP (PIB) (million RON) | 416,007 | 524,389 | 510,523 | 533,881 | 565,097 | 596,682 | 637,456 | 668,144 | 712,659 | 762,342 | 858,333 |
| IS | 0.073 | 0.083 | 0.081 | 0.057 | 0.048 | 0.056 | 0.041 | 0.036 | 0.035 | 0.036 | 0.037 |

Source: values based on financial statements of NFI and national financial accounts for 2007–2017 (Ministry of Public Finance).

The evolution of the values of this developmental indicator signifies the growth of the NFI sector from the perspective of financial assets, determined primarily by increase in GDP growth and an absolute increase in the value of financial assets of the sector. This development highlights the potential enhancement of sector participation in GDP formation. NFI development highlights the relative disconnection from efficiency and performance indicators. This indicator also reveals the effects of inadequate portfolio management and policy as well as the deterioration of the financial situation.

Indicator 4: Debt Sustainability Indicator

Formula:
$$IT = DFN/VTN = \text{total financial debt/total income}$$

Values of the debt sustainability indicator (IT) were calculated and are reported in Table 7 and Figure 2.

**Table 7.** NFI debt sustainability indicator (IT) values for the period 2007–2017.

| Year | 2007 | 2008 | 2009 | 2010 | 2011 | 2012 | 2013 | 2014 | 2015 | 2016 | 2017 |
|---|---|---|---|---|---|---|---|---|---|---|---|
| Liability (DFN) (million RON) | 48,410 | 43,502 | 27,216 | 22,527 | 12,300 | 24,189 | 2402 | 23,848 | 22,404 | 23,311 | 24,920 |
| Income (VTN) (million RON) | 2756 | 2689 | 3002 | 6206 | 5880 | 5034 | 4882 | 5168 | 4272 | 4382 | 4955 |
| IT | 17,565 | 16,177 | 9066 | 3630 | 2092 | 4805 | 4589 | 4615 | 5244 | 5319 | 5029 |

Source: values based on financial statements of NFI and national financial accounts for 2007–2017 (Ministry of Public Finance).

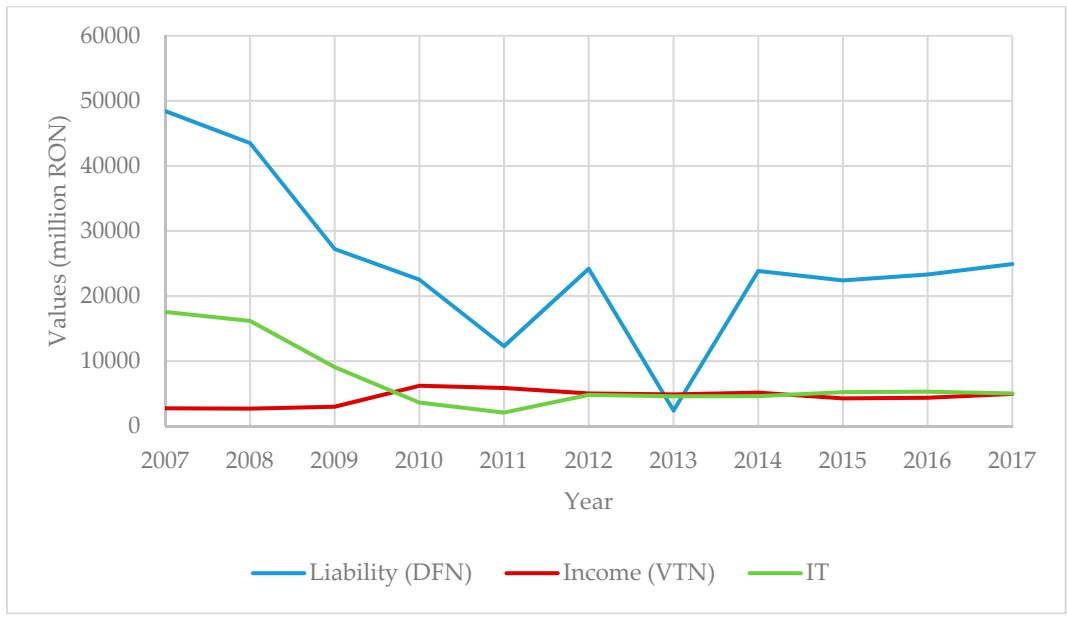

**Figure 2.** Debt sustainability values of the NFI (IT) sector.

The continuous diminution of the value of this indicator over the period 2007–2011 highlights the reduction in the capacity of financial investments to support, through total revenues generated, the commitment of financial liabilities. The insufficient capitalization of the leverage potential, determined by the increase in equity and income, reveals the inability of asset management to profitably capitalize the asset portfolio. In 2014–2017, there was an increase in the IT indicator, reflecting an increase in investment capacity. The indicator values for the full 10-year period are summarized in Table 8.

**Table 8.** Values of nonbank financial institutions' indicators for the period 2007–2017.

| No. | Dimension | Ind. | 2007 | 2008 | 2009 | 2010 | 2011 | 2012 | 2013 | 2014 | 2015 | 2016 | 2017 |
|-----|-----------|------|------|------|------|------|------|------|------|------|------|------|------|
| I. | Efficiency | IF | 0.054 | 0.055 | 0.549 | 0.198 | 0.208 | 0.178 | 0.183 | 0.198 | 0.155 | 0.139 | 0.137 |
| II. | Performance | IU | 0.092 | 0.063 | 0.083 | 0.201 | 0.216 | 0.188 | 0.188 | 0.217 | 0.169 | 0.156 | 0.153 |
| III. | Development | IS | 0.073 | 0.083 | 0.081 | 0.057 | 0.048 | 0.056 | 0.041 | 0.036 | 0.035 | 0.036 | 0.037 |
| IV. | Sustainability | IT | 17,565 | 16,177 | 9066 | 3630 | 2092 | 4805 | 4589 | 4615 | 5244 | 5319 | 5029 |

Source: values based on financial statements of NFI and national financial accounts for 2007–2017 (Ministry of Public Finance).

The evolution of the indicators of nonbank financial institutions at the national level in Romania shows us how to involve them in the financing of economic activities of nonbanking populations. Moreover, the working hypothesis by which the role of nonbank financial institutions becomes the direct source involved in the process of inclusion is confirmed by the financial indicators calculated here.

*4.2. The Microinnovation and Entrepreneurship (MIT) Model*

After three years of searches, documentation, research, forecasting, testing, and implementation, we propose a microfinance model—the MIT. This model not only meets international standards and practices but also responds directly to the needs of entrepreneurs in the rural area of Romanian.

In Romania, there are over 3.5 million small firms, peasant farms, and other types of household businesses that need access to financing. Moreover, according to the Romanian National Statistics 2014, 5.5 million people are suffering relative or absolute poverty. Our field interview has shown a strong inclination of the younger generation to start their own businesses. The population aged between 16 and 24 is 420,000. Among them, the unemployment rate is over 20%. Following the 2008 financial crisis, the banking system has considerably reduced their rural branches, making the scarcity of microfinance more acute. Therefore, it is vital for Romanian rural area economic development to form new types of financial institutions and offer innovative financial products. The following observations and considerations demonstrate the need to develop this MIT model.

The nonbanking financial sector has risen as a result of legislative interventions from 2007 to 2015. Applied research has been undertaken on microfinance in rural areas, including analyzing and testing the entrepreneurial capacity to set up microfinance institutions in rural areas. Along with the impact of financing policies on Romanian rural areas, there is also a great possibility of setting up a network of 1580 microenterprises, specialized in rural microfinance services for communes with more than 3000 inhabitants. Furthermore, past and current agricultural credit, in terms of the number of communes in Romania classified according to the number of inhabitants, creates possible direct beneficiaries of microfinance.

The MIT model, which incorporates elements from each of the microfinance models, is considered the most applicable to the rural area of Romania. The MIT model takes into account traditional instruments and products that have been transposed to beneficiaries through microfinance instruments using financial technologies. The program in this model is designed especially for the small entrepreneur from Romanian rural areas, following the test for entrepreneurial capacity, acquisition of competencies, and coordinated implementation. In addition, a network with an architecture of financial flows and stocks (transmission mechanism, flow, transmitters, and receivers) is clearly identified in the MIT model. A detailed description of the model development is beyond the scope of this paper. For further detail, see Manta (2018); however, the features of the model are summarized in Figure 3.

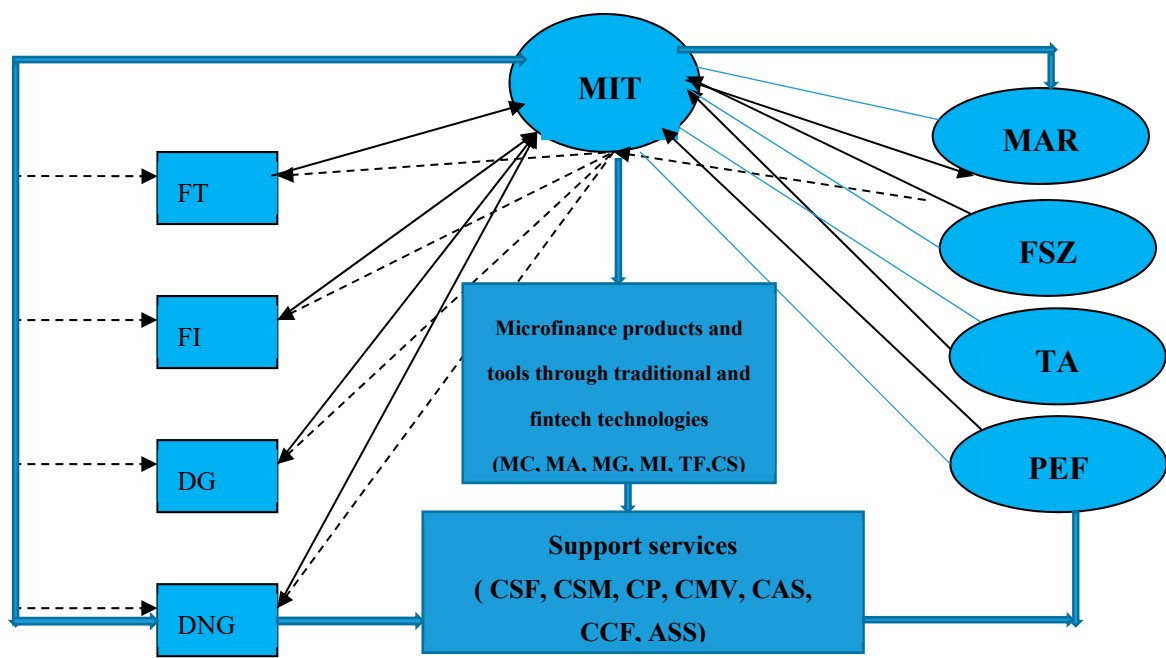

**Figure 3.** The microfinance entrepreneur model.

The emergence of new systems such as GlobTech and FinTech has stimulated confidence in new concepts of financial products from microfinance institutions (i.e., microfinance enterprises (MSMs)). The beneficiaries of microfinance products include rural entrepreneurs in rural areas, FSZ—semisubsistence agricultural farms, TA—young entrepreneurs, and financially excluded donors (the poor). The funding in this system could be sourced from (1) FT—traditional donors (banks, NFIs, international financial institutions, etc.); (2) FI—investment funds; (3) DG government donors (government, government agencies, etc.), especially in the case of programs for stimulating population lending through social credit with zero interest and having a major impact on financial support, financial education, social integration, fiscal consolidation, and sustainability in the rural environment; and (4) DNG—nongovernmental donors (international and national associations, foundations, etc.).

Microfinance products and tools related to the model were distributed to recipients through either classic channels (1580 MSM networks) and/or current digital financial platforms. From 2017 onwards, microcredit products and the transfer of funds have been the tools/products that are most easily handled by FinTech's current technology. These include

- Microcredit MC;
- MA—microinsurance;
- MG—microguarantee;
- MI—microinvestment/microeconomics;
- TF—transfer of funds;
- CS—social credit.

  Support services provided by MSM in the MIT model include

- CSF—financial consultancy;
- CSM—management consulting;
- CP—design consultancy;
- CMV—marketing and sales consultancy;
- CAS—assistance association in associative forms;
- CCF—advisory services for accounting and tax services;

- ASS—other support services.

Just as the family is the basic cell of society, the small entrepreneur (the SME and/or the microenterprise) is the basic cell of the rural economy, and this basic unit concept determined the emergence of the MIT model. The procedure for setting up an MSM is the same as for any limited liability company (LLC), commonly referred to as the specialized microfinance (MSM), SRL. Establishment of an MSM SRL company in 2017 could be done within three working days of submitting the file to the Trade Registry in the area where the company's registered office is located. Steps to set up an MSM-type firm in 2017 are listed below:

(1) Select and book the business name with the specially mentioned MSM. To save time, the entrepreneur can prepare at least three names when they check name availability. The National Trade Register Office charges a fee for the registration procedure. A special MSM SRL must include the name of the locality in the name for the identifications within the network.

(2) Establish the main activity object. The firms specializing in microfinance can use CAEN CODE—6612 Financial intermediation activities and/or other specific CAEN codes of activity as their core code. A lawyer will check all CAEN codes.

(3) Deposit social capital with the bank, which will also serve as the treasurer bank. The minimum social capital in the case of an LLC is 200 RON (equivalent 45 EUR) and must be deposited in the company's account with a commercial bank.

(4) Set up the registered office. The law requires an MSM to have an active working place. Eligible places include the company owner's personal property (sales contract or heir certificate needed as proof), rented spaces (rental or sublease agreements registered with the territorial fiscal units needed as proof), spaces with commodity contracts (commodity contract, usufruct), or leased real estate (real estate leasing contract needed as proof).

(5) Draft the company charter. Prepare the document of the specialized MSM SRL following the standards, including all the specific clauses for a limited liability company.

(6) Prepare the entrepreneur's own declaration, showing that he/she fulfills the legal conditions for having the status of associate and/or administrator and obtaining the specimen signature.

## 5. Discussion and Conclusions

Globally, an enormous expansion of NFIs has occurred to meet the demand for financial services from huge populations currently inadequately serviced. This trend has been facilitated by new technologies and international acknowledgement of the benefits of financial inclusion that participation in regulated financial systems brings. Concurrently, global investment in FinTech has also increased dramatically. Focusing on a national level, it has been demonstrated that the financial development of the NFI sector has a relevant impact on the long-term performance and growth of the economy, measured by factors such as the size, depth, access, efficiency, and stability of the NFI sector and the financial system. As part of this development, we have proposed the network model of NFIs, based on the indicators calculated by the authors, whereas the new microfinance model is oriented towards improving the social performance of the entrepreneurs in rural Romania and is explained in terms of defined assumptions, identified needs, model components, and inter-relationships. A short, simple entry process to the package is also listed.

The evaluation of the development of NFIs was made by indicators that express the weight of the financial potential (assets) of this sector in the macroeconomic or macrofinancial situation and the financial structure of the sector or institutions. By calculating indicators of the financial nonbanking sector, this study has demonstrated the size of the NFI sector in Romania and indicated an early stage of development with considerable potential. It is emphasized that financial system development is a process of consolidation and diversification of NFIs to provide services that meet the specific requirements of disparate customers (or economic units) in an effective and real way. Once established, the provision of uninterrupted and unlimited services must be ensured by relevant institutions.

Social and economic sustainability requires the creation of a social system that supports the objectives of raising real incomes, raising educational standards, and improving health and the quality of life. If development is restricted by resources, the priority should involve renewable natural resources, as well as respecting the limits of the development process, allowing for the fact that those limits may be adjusted by technology. The financial sustainability of the NFI sector is achieved when the levels and standards of NFI services are provided in line with long-term objectives, without increasing customer payments or reducing the quality of services.

Therefore, our work contributes to the scientific results by establishing network operating mechanisms for financial institutions involved in supporting social and financial inclusion, as well as microfinance models designed and adapted to local specificities but structured based on existing empirical research at a global level. In addition, our study presents an applicative direction, since the microfinance business model for Romanian rural entrepreneurs is currently testable at the level of NFIs. From an institutional decision point of view, the proposed network contributes to the financial inclusion of small entrepreneurs in rural areas of Romania. Furthermore, it can be used as a supporting instrument for the process of regulating financial services provided to beneficiaries through NFIs. In our future research, we will gather concrete results of these effects that microfinance models have for entrepreneurs. We will also propose the development of innovative mechanisms and tools for the social and financial inclusion of nonbanking institutions (small entrepreneurs) in the context of financial technologies, with a direct impact on the sustainability of small businesses in rural areas.

In summary, in many rural areas where people are excluded financially, the role of the financial network of NFIs is very important, with both social and financial impacts. Both banking and nonbanking institutes should look at financial inclusion both as a business opportunity and as a social responsibility. This research has made a scientific contribution to financial models, providing indicators that could be used within financial banking institutions and a pragmatic package oriented to microfinance solutions for entrepreneurs locally. This study can be extended in future research to present the global evolution of both the financial and social inclusion processes, as well as the implications for financial institutions that support this process.

**Author Contributions:** Conceptualization, O.M., X.-F.S., and X.-G.Y.; methodology, Y.C.; software, O.M.; validation, X.-G.Y. and N.D.; writing—original draft preparation, O.M. and X.-G.Y.; writing—review and editing, X.-F.S. and X.-G.Y.; funding acquisition, O.M.

**Funding:** This research received no external funding.

**Acknowledgments:** We deeply appreciate the three anonymous reviewers for their intuitive suggestions and constructive comments. We are grateful to the editors for their continued support and effort for our manuscript.

**Conflicts of Interest:** The authors declare no conflict of interest.

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
