# Peer review of "Social and Financial Inclusion through Nonbanking Institutions: A Model for Rural Romania"

_jrfm, doi:10.3390/jrfm12040166_

Round 1
Reviewer 1 Report
attached

Author Response
Dear Professor Dr.,
First, we would like to thank you for your effort and support in reviewing our work. We have fully considered all suggestions and recommendations, and revised the manuscript accordingly. Changes are summarized below.
Title: has been modified to now include reference to Romania – as suggested.
Abstract: this has been extensively re-drafted and refined – to be more consistent with the required structure, while remaining within stated word limits. The focus is now on the Romanian site and model, as well as the practical implications for rural populations in developing countries everywhere.
1/2. Introduction: slightly shortened, but revised in a way that (in combination with the Literature Review ) makes the aims and conclusions of the work more explicit.
Methodology: small changes have been made to improve readability where possible; however, if there are further specific suggestions for improved presentation we are prepared to consider them.
Results and Discussion: minor text changes have been incorporated to improve readability and flow, together with slightly elaborated explanation of Table values. Some minor modification / standardization of Tables and Figures also undertaken.
Conclusions: this has been modified to remove general background, while emphasizing the key findings and broader implications.
References: all text citations re-checked – list entries standardized to journal format, with more URL detail added where necessary.
Please find attached our revised manuscript!
We hope that these changes are considered satisfactory responses to reviewer recommendations.
I wish you all the best,
Kind regards,
Xiao-Guang Yue , Yong Cao, Nelson Duarte , Xue-Feng Shao , Otilia Manta
Reviewer 2 Report
The authors discuss the social challenges in financial systems which have both an immediate and medium-long-term. They discuss the new models and solutions to these challenges, especially for many individuals with little or no access to financial services.
They note that while current communication technologies make these services more accessible, through mobile digital platforms, there are still difficulties in establishing workable customer arrangements. They note that complementary organizations, non-bank financial institutions have started to address this situation by offering more flexible services, tailored to individual circumstances.
I believe that the title, abstract and the introduction is a little misleading in the sense that it does not indicate that this is a study on a rural area in Romania. You need to read through a chunk of the article to get to this.
Moreover, maybe the authors could elaborate further on the importance of such a study and for which societies? elaborate further on the sustainability of whom and where?
Author Response
Dear Professor/Dr.,
First, we would like to thank you for your effort and support in reviewing our work. We have fully considered all suggestions and recommendations, and revised the manuscript accordingly. Changes are summarized below.
Title: has been modified to now include reference to Romania – as suggested.
Abstract: this has been extensively re-drafted and refined – to be more consistent with the required structure, while remaining within stated word limits. The focus is now on the Romanian site and model, as well as the practical implications for rural populations in developing countries everywhere.
1/2. Introduction: slightly shortened, but revised in a way that (in combination with the Literature Review ) makes the aims and conclusions of the work more explicit.
Methodology: small changes have been made to improve readability where possible; however, if there are further specific suggestions for improved presentation we are prepared to consider them.
Results and Discussion: minor text changes have been incorporated to improve readability and flow, together with slightly elaborated explanation of Table values. Some minor modification / standardization of Tables and Figures also undertaken.
Conclusions: this has been modified to remove general background, while emphasizing the key findings and broader implications.
References: all text citations re-checked – list entries standardized to journal format, with more URL detail added where necessary.
Please find attached our revised manuscript!
We hope that these changes are considered satisfactory responses to reviewer recommendations.
I wish you all the best,
Kind regards,
Xiao-Guang Yue , Yong Cao, Nelson Duarte , Xue-Feng Shao , Otilia Manta
Round 2
Reviewer 1 Report
I don't see the paper significantly improved. Please refer to my previous report and provide point by point responses.
Author Response
Thank you for your comments. Please kindly find the correct responses as attachment.

Round 3
Reviewer 1 Report
Need to point out future research directions according to my previous referee report. Cover more literature and you can draw more attention.
Author Response
Dear Professor,
We would like to thank you for your advice and guidance. Specific responses to your comments are listed in the sections below:
Need to point out future research directions according to my previous referee report. Cover more literature and you can draw more attention.
We pointed out the future research directions according to the previous referee report. We added a new paragraph in the result section and we also rewrote the discussion and conclusion section. Moreover, we added two articles you suggested and eight other new peer-reviewed journal articles which are relevant to this topic and contribute the literature.
Many special thanks for all your support!
All the best,
Xiao-Guang Yue, Yong Cao , Nelson Duarte , Xue-Feng Shao, Otilia Manta